# EGFR Transgene Stimulates Spontaneous Formation of MCF7 Breast Cancer Cells Spheroids with Partly Loss of HER3 Receptor

**DOI:** 10.3390/ijms222312937

**Published:** 2021-11-29

**Authors:** Olga Troitskaya, Diana Novak, Anna Nushtaeva, Maria Savinkova, Mikhail Varlamov, Mikhail Ermakov, Vladimir Richter, Olga Koval

**Affiliations:** 1Institute of Chemical Biology and Fundamental Medicine SB RAS, 630090 Novosibirsk, Russia; nushtaeva.anna@gmail.com (A.N.); m.savinkova@g.nsu.ru (M.S.); mvarlamov@gmail.com (M.V.); ermakovm97@gmail.com (M.E.); richter@niboch.nsc.ru (V.R.); o_koval@ngs.ru (O.K.); 2Department of Natural Sciences, Novosibirsk State University, 630090 Novosibirsk, Russia; d.novak@g.nsu.ru

**Keywords:** 3D cell culture, spheroids, EGFR, HER3, MCF7, cancer stem cells, drug resistance

## Abstract

Multicellular spheroids with 3D cell–cell interactions are a useful model to simulate the growth conditions of cancer. There is evidence that in tumor spheroids, the expression of various essential molecules is changed compared to the adherent form of cell cultures. These changes include growth factor receptors and ABC transporters and result in the enhanced invasiveness of the cells and drug resistance. It is known that breast adenocarcinoma MCF7 cells can spontaneously form 3D spheroids and such spheroids are characterized by high expression of EGFR/HER2, while the natural phenotype of MCF7 cells is EGFR^low^/HER2^low^. Therefore, it was interesting to reveal if high epidermal growth factor receptor (EGFR) expression is sufficient for the conversion of adherent MCF7 to spheroids. In this study, an MCF7 cell line with high expression of EGFR was engineered using the retroviral transduction method. These MCF7-EGFR cells assembled in spheroids very quickly and grew predominantly as a 3D suspension culture with no special plates, scaffolds, growth supplements, or exogenous matrixes. These spheroids were characterized by a rounded shape with a well-defined external border and 100 µM median diameter. The sphere-forming ability of MCF7-EGFR cells was up to 5 times stronger than in MCF7^wt^ cells. Thus, high EGFR expression was the initiation factor of conversion of adherent MCF7^wt^ cells to spheroids. MCF7-EGFR spheroids were enriched by the cells with a cancer stem cell (CSC) phenotype CD24^−/low^/CD44^−^ in comparison with parental MCF7^wt^ cells and MCF7-EGFR adhesive cells. We suppose that these properties of MCF7-EGFR spheroids originate from the typical features of parental MCF7 cells. We showed the decreasing of HER3 receptors in MCF7-EGFR spheroids compared to that in MCF^wt^ and in adherent MCF7-EGFR cells, and the same decrease was observed in the MCF7^wt^ spheroids growing under the growth factors stimulation. To summarize, the expression of EGFR transgene in MCF7 cells stimulates rapid spheroids formation; these spheroids are enriched by CSC-like CD24^−^/CD44^−^ cells, they partly lose HER3 receptors, and are characterized by a lower potency in drug resistance pomp activation compared to MCF7^wt^. These MCF7-EGFR spheroids are a useful cancer model for the development of anticancer drugs, including EGFR-targeted therapeutics.

## 1. Introduction

Breast cancer is the leading form of cancer in woman worldwide with 471.5 cases per 100,000 population in Russian Federation [1]. A wide range of drugs is available for breast cancer therapy, including targeted drugs, but the death rate of advanced breast cancer patients remains high. Traditional models for screening antitumor agents are 2D monolayer cell models, which lack many of the important features of a tumor in the body-gradient of nutrients and metabolites, the gradient of pH and oxygen, as well as interactions of cancer cells with an extracellular matrix. 3D multicellular cultures overcome these disadvantages [2]. 3D cultures are widely used in investigations of cell–cell interactions, cell differentiation, drug delivery, and their efficiency and toxicity [3,4,5,6]. Moreover, the ability of tumor cells to spontaneously form spheroids into the blood circulation from solid tumors is believed to play a role in metastasis [7].

For the various types of tumor cells, the potential of spheroid formation is different: in most cases, stable 3D models require specific culturing plates and additional matrix or growth factors, all of which raise the price and effort of the research [8]. Therefore, cell lines capable of efficiently and reproducibly forming 3D models under routine cultivation conditions may be useful for the molecular biologist.

To verify the suitability of a particular 3D cell model for a study, it is necessary to determine the similarities and differences between the original 2D and 3D cultures. Stem-like cell population within tumors is a source of recurrence and metastasis of breast cancer [9,10,11]. These cells are known as cancer stem cells (CSCs) because of their ability for self-renewal and differentiation into various different cancer cells. CD44 and CD24 are specific markers of CSC cells from breast tumors [12]. For various types of neoplasia, it has been shown that spheroid formation can increase the content of CSC cells or cells with stem cell-related characteristics in culture [13]. The proportion of CSC cells in culture can influence sensitivity to various antitumor drugs. Often it is the CSC cell population that determines the development of drug resistance through the activation of ABC transporters such as P-glycoprotein (P-gp) [14]. Effective targeting of ABC efflux pumps is important to make CSCs penetrable to chemotherapeutics [15].

The MCF7 cell line is one of the most demanded systems of breast cancer for the investigation of tumor progression and anticancer therapeutics analysis. These cells able to produce spheroids under the standard conditions of cultivation with low efficiency and poor replicability. For reproductive spheroid formation for MCF7 cells, Kelm et al. described a hanging drop method [16]. In some cases, the potential of multicellular spheroid formation positively correlates with the basal EGFR expression levels [17]. EGFR is a member of the ErbB family of receptor tyrosine kinases which also include ErbB2/Her2, ErbB3/Her3, and ErbB4/Her4 [18]. MCF7 cells have been described as relatively low in EGFR expression among breast cancer cell lines [19]. Kim et al. demonstrated that pharmacological blockade of EGFR significantly suppressed sphere formation in MCF7^wt^ cells [20]. All these data point to the importance of EGFR expression for spheroid formation of MCF7 cells.

In the present work, MCF7 with hyperexpression of EGFR was obtained. Two main questions were posed by the study: (i) whether high EGFR expression is sufficient to transform adhesive MCF7 cells to 3D culture, and (ii) how the transformation into spheroids changes cellular markers associated with tumor progression and drug resistance in MCF7-EGFR spheroids.

## 2. Results

### 2.1. MCF7-EGFR Cell Cultivation and Spheroid Formation

MCF7 cell line with high expression of epidermal growth factor receptor (EGFR) was constructed using the retroviral transduction method. The resulting MCF7-EGFR cell line was obtained after FACS separation of the cell population with a high EGFR signal from puromycin-resistant cells (Figure 1).

These cells were seeded into the adhesive 24-wells plates (Eppendorf™ plates) 800 cells/well and cultured under standard conditions. MCF7-EGFR cells assembled in spheroids very quickly and grew predominantly as a 3D suspension culture with no special coating plastic, growth supplements, or additional matrixes (Figure 2A–E). MCF7 cells, transduced by the control retroviral particles, did not produce any spheres during four passages after transduction.

The curves of multiplication of spheroid number during nine days demonstrated the rapid initiation of spheroid formation in MCF7-EGFR cells (Figure 2I). The method of spheroid cultivation is summarized in Figure 3. The essential step of spheroid formation was the pre-forming of a small adhesive islet from where the sphere-initiating cells split off (Figure 2D and Figure 3 step II).

Next, these spheroids were grown as floating spheres. Spheroids are characterized by a rounded shape, a median diameter 100 µM, and a well-defined internal border. To measure sphere-forming ability, we counted spheroids in each well.

The trend of the later stages of cultivation was to increase the size of the spheroids already formed instead of increasing the number of spheroids (Figure 2D,E). For such large spheroids, histological analysis was made (Figure 2F–H). The voids observed most likely correspond to the necrotic core of the spheroid. The presence of junction organizing zona occludens protein 1 (ZO-1) and tight junction marker E-cadherin in spheroids were confirmed via immunofluorescent staining (Figure 4).

The relocation of spheroids into special low-adhesive plates greatly accelerated the increase in the number of spheroids only in MCF7-EGFR cells. These data showed a high sphere-forming ability of MCF7-EGFR cells. Moreover, step II from Figure 3 morphologically resembles the exit of individual cells from the tumor node, movement, adhesion in a new site, growth of the node, and further detachment of individual cells from this node, i.e., the process of metastasis and invasion.

To estimate the growth potential of parental MCF7 and MCF7-EGFR cells under EGFR-inhibiting conditions, EGFR inhibitor tyrphostin (AG1478) was added to the cells [21]. MTT analysis showed that MCF7-EGFR cells did not become more sensitive to EGFR inhibitor: for MCF7^wt^ and MCF7-EGFR IC50 = 13 ± 5 µM and IC50 = 17 ± 2 µM, respectively (Appendix A). This resistance could be due to the parallel activation of the drug pumps of multidrug resistance proteins (MDR). To study the effect of the EGFR inhibitor on spheroid growth, we used a concentration of AG1478 of 10 µM. We found that AG1478 had no suppressive effect on spheroid formation (Figure 2F).

To ensure that high EGFR expression was sufficient to transform the adhesive cell culture to spheroids, we constructed the similar EGFR high cells of human embryonic kidney cells HEK293. HEK293 cells grow as an adherent culture and have no or low basal EGFR, thus allowing us to evaluate phenotype changes in response to EGFR transgene overexpression. These HEK293-EGFR cells were engineered by a similar technique as MCF7-EGFR cells were (Appendix A). We did not find any 3D structures in HEK293 cells under standard conditions of cultivation when MCF7-EGFR cells efficiently form spheroids. The findings indicate that the sphere-forming ability of MCF7-EGFR originated from the typical features of parental MCF7 cells. The expression of exogenous EGFR acts as an initiating factor that triggers molecular cascades leading to less adherent ability of MCF7-EGFR cells.

### 2.2. CD24/CD44 Content in MCF7-EGFR Spheroids

We compared the representation of CD44 and CD24 markers on MCF7^wt^ cells, MCF7-EGFR adherent cells, and MCF7-EGFR spheroids. EGFR transgene did not stimulate the increase of CD24^−/low^/CD44^+^ stem-like cells population (Figure 5).

The content of these CD24^−/low^/CD44^+^ cells was approximately equal in the parental MCF7^wt^ cells and spheroids. The main difference concerned the CD24^−/low^/CD44^−^ population, which appeared in spheroids (Figure 5A, quadrant Q1). Moreover, the increase in CD24^−/low^/CD44^−^ in spheroids was accompanied by the decrease of CD24^+^/CD44^+^ cells (Figure 5B). A number of studies describe CD24^−/low^/CD44^−^ cells as a population with stem plasticity which can realize transition into CD24^−/low^/CD44^+^ cells.

### 2.3. Sphere-Forming Ability Leads to the Partial Loss of HER3 Receptor

EGFR (ErbB1) belongs to the ErbB receptor’s family, where two more therapeutically relevant receptors, HER2 (ErbB2) and HER3 (ErbB3), are included. We compared HER2 and HER3 content in MCF7^wt^, MCF7-EGFR adherent cells and in MCF7-EGFR spheroids. MCF7-EGFR adherent cells were obtained after the sorting stage as cells attached to the substrate. MCF7^wt^ cells were HER3 positive. MCF7-EGFR spheroids contained a portion of HER3 positive and a portion of HER3 negative cells. These data indicated that part of MCF7-EGFR spheroids had lost HER3 receptors in comparison with MCF7^wt^ (Figure 6A, quadrant Q2).

Moreover, MCF7-EGFR spheroids completely lost their HER3^bright^ subpopulation (Appendix A). To understand if the transformation from adhesive state to spheroids leads to the loss of HER3 we added a sphere-promoting growth factors cocktail to the MCF7^wt^ cells [22]. As already shown (Figure 2I), the formation of MCF7^wt^ spheroids was a very slow process, which led to the accumulation of spheroids with activated necrosis and senescence in a core area. Given such a limitation, stimulation with growth factors was a necessary step to obtain well-proliferated spheroids of MCF7^wt^ in a sufficient amount for analysis. Cultivation of MCF7^wt^ cells with growth factors cocktail resulted in MCF7^wt^ 3D system as soon as four days after the growth factors adding. The analysis of HER3 receptors in MCF7^wt^ spheroids demonstrated the same decrease of HER3-positive cells as that in MCF7-EGFR spheroids.

In addition to HER3, MCF7-EGFR spheroids have partly lost the HER2 receptor (Figure 6B). To summarize, MCF7-EGFR spheroid cells were depleted in both HER2/HER3 receptors.

### 2.4. Rhodamine 123 Efflux

Mobile cancer cells capable of migration and metastasis often show resistance to antitumor drugs through activation of transmembrane drug pumps such as P-gp. Multiple drug resistance is also attributed to CSC. To compare P-gp drug efflux in MCF7^wt^, MCF7-EGFR adherent cells, and MCF7-EGFR spheroids, the Rhodamine 123 (Rh123) efflux test was performed. Cells were incubated with Rh123 for 45 min, and after that, the intracellular Rh123 content was assayed by flow cytometry. Time-dependent curves of cellular Rh123 content showed that the major portion of Rh123 was eliminated from the cells for an initial 30 min (Figure 7A). In order to confirm the specificity of the efflux, P-gp inhibitor verapamil was used. We found that verapamil partly inhibits rhodamine 123 efflux in MCF^wt^ and MCF-EGFR cells but not in MCF-EGFR-spheroids (Figure 7B).

The rate of Rh123 efflux was approximately the same for all three cell lines, with a tendency to increase in MCF7-EGFR cells.

The sensitivity of parental MCF7, MCF7-EGFR cells, and MCF7-EGFR spheres to cytostatic drugs was investigated using cisplatin. Figure 7C showed that MCF7-EGFR cells were more resistant to cisplatin than MCF7^wt^ cells. Taking into account the tendency to raise Rh123 efflux in MCF7-EGFR cells, their low sensitivity to EGFR inhibitor AG1478, and low sensitivity to cisplatin, we can conclude that MCF7-EGFR have acquired drug resistance. The highest cisplatin IC50 value calculated for spheroids may be the result of delayed diffusion of the drug to the cells of the inner spheroid area.

## 3. Discussion

Anticancer drug discovery and their pharmacological applications dictate needs in cellular models that are closer to in vivo conditions. More realistic cell-to-cell interaction is realized in 3D models, and such systems have the potential to greatly improve cell-based drug screening [6]. The ability of cells to self-assemble is not extraordinary itself, but usual to culturing the cells in 3D models a special systems or supplements, e.g., hanging drop, scaffolds and hydrogels, special plastic, and cocktails of growth factors are needed [2,23]. Various methods have been described for obtaining 3D MCF7 cell culture [20,24,25,26]. Depending on the technology used, the basic molecular characteristics of the cells in 3D may vary, which can result in difficulties in reproducing such models. Therefore, the development of a simple and reproducible method of producing 3D cultures is an urgent task.

EGFR is overexpressed in many tumors of epithelial origin [27,28]. Approximately half of the cases of triple-negative breast cancer and inflammatory breast cancer overexpress EGFR, but MCF7 cells, a model of hormone-positive breast cancer, are characterized as EGFR^low^ cells [29]. Braunholz et al. found that EGFR activation was a crucial factor of anchorage-independent cultivation of head and neck squamous carcinoma cells [17]. Thus, we can formulate the paradox of MCF7 cells that makes them special from others: they are cells with low EGFR expression and a pronounced ability to spontaneous spheroid formation. Here we constructed EGFR-overexpressed MCF7 cells, which grew up predominantly as non-adhesive spheres with no special supplements, scaffolds, etc. Tight junctions between the cells in these spheres were maintained by intercellular junction proteins ZO-1 and E-cadherin as judged by immunostaining. Tight junctions are intercellular junctions critical for building the epithelial barrier and maintaining epithelial polarity [30].

The ability to form multicellular 3D spheres in vitro when grown in nonadherent serum-free conditions has been shown for cells with CSCs phenotype [31]. Bahmad et al. have developed a sphere-formation assay (SFA) for in vitro analysis of the presence of CSCs in different prostate cancer models [23]. These studies confirm the relationship between the ability of cells to form spheroids and the CSC content of cell culture. Nevertheless, CSC content in MCF7 is much lower than in MDA-MB-231 triple-negative breast cancer cells with a weak potency to spontaneous sphere formation [32]. In our work, we did not reveal the changes in CD44^+^/CD24^low/−^ CSC cells number among MCF7^wt^ and MCF7-EGFR spheroids. These data do not negate the higher sphere-forming ability of cells with the CD44^+^/CD24^low/−^ phenotype, but in our work, it should be taken into account that the primary selection of MCF7-EGFR cells was performed by the expression level of exogenous EGFR, and two populations among transformed cells: CD44^+^/CD24^low/−^ population and EGFR^high^ population may not have been matched. Thus, in our case, EGFR expression was a more important characteristic for the transition of adherent MCF7 to 3D than the enrichment of CSC-like cells. The expression of EGFR transgene acts as an initiating factor that triggers molecular cascades leading to less adherent ability of MCF7-EGFR cells.

We have found that MCF7-EGFR spheroids were enriched by CD24^−^/CD44^−^ cells (up to 12%), which are another CSC-related phenotype in comparison to MCF7^wt^ (up to 3.5%). Recently, Qiao et al. showed that a high frequency of CD44^−^/CD24^−^ cells is associated with delayed postoperative breast cancer metastasis [33]. They demonstrated that under certain conditions, CD44^−^/CD24^−^ is able to spontaneously transform into CD44^+^/CD24^−^ CSC cells. Litviakov et al. has found that dedifferentiation of CD44^low^ to CD44^high^ breast cancer cells is linked to stemness genes amplification [34]. Yan et al. have examined MCF7 cells’ distribution into different populations in terms of CD44/CD24 marker content [35]. They showed that cells with CD44^−^/CD24^−^ and CD44^−^/CD24^+^ phenotypes did not form tumors when transplanted into mice. In general, cells in the core of spheroids grow under higher hypoxic conditions than adherent cancer cells resulted in the activation of CSC phenotype. Denes et al. found a positive correlation between stem cell presence in multicellular spheroids and hypoxia [7]. Thus, the lack of enrichment by the CSC population in MCF7-EGFR spheroids distinguishes these spheroids from other similar 3D systems.

On the other hand, CD44 should be considered not only as a stemness marker but also as an adhesion protein. Of note, CD44 is a major cell surface receptor for hyaluronan, and it also interacts with osteopontin, collagen, and laminin [36]. Because the most significant difference between MCF7-EGFR cells and MCF7-EGFR spheroids was the decrease of CD44 receptors (from an average of 94 to 67%, respectively; see Figure 4), one would expect a deterioration of adhesive properties of the cells with low CD44 levels.

We have found that cells from MCF7-EGFR spheroids have partly lost HER3 receptors. Previously, the same decrease of HER3 we have observed in primary breast cancer cells after rounds of hypoxic conditions (pulsed hypoxia) [37]. As spheroids derived from MCF7^wt^ cells after stimulation with growth factors also lost HER3, it can be assumed that poor oxygenation within the spheroids may negatively regulate HER3 expression.

Cancer hypoxia can promote a more aggressive phenotype with therapeutic resistance. Chun and co-authors have been demonstrated that spontaneous spheroids of MCF7 cells contains high proportion of MDR-1 positive cells with resistance to various conventional antitumor therapeutics [38]. Surprisingly, in the case of MCF7-EGFR spheroids, the activity of the drug resistance pump did not increase than in parental MCF7^wt^ cells. We can speculate that such features of MCF7-EGFR spheroids are related to the fact that there was no increase in the population of CSC-like cells.

## 4. Materials and Methods

### 4.1. Cell Lines 

Human adenocarcinoma cells MCF-7 (purchased: #ACC 115, DSMZ, Braunschweig Germany) were maintained in Iscove’s modified Dulbecco’s medium (Sigma-Aldrich, Burlington, MA, USA), supplemented with 10% heat-inactivated fetal calf serum and 1% penicillin-streptomycin in 5% CO_2_.

HEK293 human embryo kidney cells were grown in DMEM: Nutrient Mixture F-12 (DMEM:F12; Sigma-Aldrich, Burlington, MA, USA) supplemented with 4 mM l-glutamine and 10% fetal bovine serum (HyClone, Logan, UT, USA), 100 U/mL penicillin, 100 µg/mL streptomycin, 1 mM sodium pyruvate, and 1× MEM non-essential amino acids. Cells were maintained in a monolayer culture as previously described [39].

### 4.2. MCF7-EGFR Cells Construction

MCF7 cells were transfected with a mixture of packaging retroviral plasmids pCMV-VSV-G (cat. No. 8454, Addgene, Watertown, MA, USA), pUMVC (cat. No. 8449, Addgene, Watertown, MA, USA), and EGFRwt transfer plasmid (cat. No. 11011, Addgene, Watertown, MA, USA), or control plasmid (cat. No. 27490; Addgene, Watertown, MA, USA) using the calcium-phosphate transfection protocol [40]. Two days later, supernatants of conditioned media were filtered using 0.45 µm PES filters and used for cell transduction either immediately or following ultracentrifugation. MCF7 cells were transduced by target or control retroviral particles using spinoculation protocol [41] in the presence of polybrene (8 µg/mL). The transduced cells were then supplied with puromycin (1 µg/mL) 2 days after transduction to discard the non-transduced cells. EGFR expression was analyzed by flow cytometry using anti-EGFR antibodies 1:100 (#MA5-13319; Invitrogen, Rockford, IL, USA), specified to extracellular domain of EGFR receptor. MCF7 cells with EGFR-high phenotype were sorted (SONY SH800S sorter, San Jose, CA, USA, 100 µm nozzle chip) into adhesive 24-wells cell culture plate (Eppendorf^TM^; Hamburg, Germany) and cultivated under standard conditions.

### 4.3. Spheroids Formation and Counting

Unless otherwise specified, MCF7^wt^ and MCF7-EGFR cells were cultured under standard conditions on the adhesive 96-wells plates (TPP, Trasadingen, Switzerland) to form spheroids. If necessary, the spheroids were transferred to low-adhesion plates (non-treated Eppendorf™ plates, #EP003730011-80EA, Hamburg, Germany) without changing the composition of the culture medium. To plot the kinetic curves of spheroid formation, the cells were gently dissociated by Stempro™ Accutase™ cell dissociation reagent (A1110501, Gibco, New York, NY, USA), seeded at a rate of 3 × 10^3^ cells/well in 96-well plates, and cultured under standard conditions as described above. The spheroids were counted daily in six independent wells of a 96-well plate using an inverted microscope (Eclipse Ti, Nikon, Tokio, Japan) at 10× magnification. In the light field, all free-floating spheroids were counted. Those spheroids whose size was smaller than 30–50 µm were not included in the total score of spheroids, and the mean value of spheroids per well and SD were calculated.

For assessing the effects of growth factors, cells were cultivated in DMEM/F12 supplemented with 1× GlutaMAX ™ Supplement (35050-061, Gibco™, New York, NY, USA), 1× Antibiotic-Antimycotic (15240-062, Gibco™, New York, NY, USA), 20 ng/mL EGF (Epidermal Growth Factor; E9644, Sigma-Aldrich, Burlington, MA, USA), 20 ng/mL fibroblast growth factor basic (bFGF, PHG0261, Gibco™, New York, NY, USA), 5 µg/mL insulin (I9278, Sigma-Aldrich, Burlington, MA, USA), 2% B27™ Plus Supplement (A35828010, Gibco™, Burlington, MA, USA), and 4% Albumin Bovine Serum Fraction V (BSA, 126593, Sigma-Aldrich, Burlington, MA, USA). This growth factor cocktail was composed according to Xie et al.’s recommendations [22]. To assess the growth parameters in non-adhesive plates, cells were seeded in Nunclon™ Sphera™ Dishes (174943, Thermo Scientific™, Waltham, MA, USA).

### 4.4. Immunocytochemistry of Spheroids

Immunocytochemistry of spheroids was performed essentially as described in [42], with some modifications. Briefly, spheroids were collected by centrifugation, washed in cold PBS, transferred onto the slide, and gently flattened under a coverslip to expose the interior. The coverslip was removed and the area with spheroids was circled with a hydrophobic marker PAP Pen (Diagnostic biosystems; Hague, The Netherlands), and flattened spheroids were fixed with 80% methanol at −20 °C within 20 min. The cells were washed twice with PBS and blocked with 5% BSA in PBS for 30 min at room temperature. The following antibodies were used for staining: anti-ZO1 tight junction protein antibody (1:100, ab216880, Abcam, Cambridge, UK), Alexa Fluor^®^ 647 conjugated donkey anti-rabbit IgG H&L (ab150075, Abcam, Cambridge, UK), Alexa488-conjugated rat anti-CD324 (E-cadherin, Invitrogen, Waltham, MA, USA) (1:100, 53-3249-80, eBioscience, San Diego, CA, USA). Samples were incubated with antibodies at 4 °C overnight and with conjugated secondary antibodies for 1 h at room temperature. Samples were also incubated with the secondary antibody alone as the isotype control. Cells were mounted in Prolong Diamond Antifade Mountant with DAPI (Invitrogen, Eugene, OR, USA) to visualize the nuclei. The slides were visualized, and fluorescence images were obtained using a fluorescence microscope Image.A2.Zeizz (Carl Zeizz, Jena, Germany) (40× magnification), Zen software version 3.0 SR.

### 4.5. Histological Analysis of Spheroids

Free-floating spheres were fixed with 10% buffered formalin (Biovitrum, Saint-Petersburg, Russia) for 20 min at room temperature, then post-fixed for 24 h at 4 °C with 30% sucrose in H_2_O. Next, spheres were placed in conical molds, poured with embedding medium for cryotomy (Tissue-Tek, Torrance, CA, USA) into blocks, and frozen at −70 °C. Obtained blocks were sliced on cryostat LEICA CM1850 UV (Leica, Wetzlar, Germany). Tissue slides were fixed with a mixture of methanol and acetone (1:1), then stained with Mayer’s hematoxylin and eosin for 3 min each. Stained cells were visualized using an Axioscop 2 PLUS fluorescence microscope (Carl Zeiss, Jena, Germany).

### 4.6. Cell Killing Assay

Cell viability was detected 72 h after drug treatment using the MTT test as in Ref. [39]. The IC50 values were calculated with CompuSyn software version 1.0.

A stock solution of 10 mM AG1478 (Sigma-Aldrich, Burlington, MA, USA) in dimethylsulphoxide (DMSO) was kept at −70 °C. For cellular experiments, AG1478 was diluted in IMDM containing 10% FBS, so the highest concentration of DMSO in cultivating wells was 0.1%. Stock solution of 0.5 mg/mL cisplatin (Veropharm, Volginsky, Russia) was kept at +4 °C.

### 4.7. Western Blot

Cells were lysed with cell lysis buffer (50 mM Tris, pH 8.0, 5 mM EDTA, 150 mM NaCl) containing 0.1% SDS, 1× complete protease inhibitor cocktail (Roche Diagnostics GmbH, Mannheim, Germany) and 1 mM PMSF. Samples (20 µg) were separated by 10% SDS-PAGE and transferred to a Trans-Blot nitrocellulose membrane (Bio-Rad Laboratories, Hercules, CA, USA) by a wet blotting procedure (100 V, 500 mA, 90 min, 15 °C) using “Mighty Small Transphor” (GE healthcare Bio-Science AB, Helsinki, Finland). Immunodetection was performed using the iBind system (Life Technologies, Waltham, MA, USA), iBind Cards (Invitrogen, Waltham, MA, USA), and antibodies. The membrane was incubated with antibodies: anti-β-Tubulin produced in mouse 1:2000 (#T8328, Sigma-Aldrich, Burlington, MA, USA) and anti-EGFR 1:1000 (#sc-373746, Santa Cruz Biotechnology, Santa Cruz, CA, USA) overnight at 4 °C in PBS. Next, the membrane was incubated with HRP-conjugated antibodies-anti-goat IgG HRP 1:1000 (#HAF109, R&D Systems, Minneapolis, MA, USA) for 1 h at room temperature in 5% non-fat dried milk. Chemiluminescence signal was produced with Novex ECL HRP chemiluminescent substrate reagent kit (Invitrogen, Waltham, MA, USA) and detected with a C-DiGit blot scanner (Li-COR Bioscience, Nebraska, NE, USA).

### 4.8. Flow Cytometry

All analyses were performed using a FACSCantoII flow cytometer (BD Biosciences, Franklin Lakes, NJ, USA), and the data were analyzed by FACSDiva Software (BD Biosciences, NJ, USA). Cells were initially gated based on forward scatter versus side scatter to exclude small debris, and ten thousand events from this population were collected. The following antibodies were used for analysis: anti-HER2-FITC anti-HER3-APC from Sony (# 162686 and # 162652, respectively), anti-CD44-APC and anti-CD24-PECy7 from BD Pharmigen (#560890 and #561646, respectively) (Sony, San Jose, CA, USA).

### 4.9. Rhodamine 123 Efflux

Rhodamine123 efflux assay was performed essentially as described in [43], with some modifications. Briefly, cells were cultured under standard conditions 48 h, after that cells (2.5 × 10^5^) were harvested by Accutase™ (Gibco, New York, NY, USA), resuspended in complete medium, and 8 mM Rh123 (Santa Cruz Biotechnology, Heidelberg, Germany) was added to cells. After 50 min of incubation with Rh123 on ice, medium was replaced to remove excess Rh123 with Rh123-free medium and cells were incubated at 37 °C for the desired time period, washed by a portion of Efflux Buffer (PBS, 0.01% sodium azide, 0.5% BSA) and analyzed by flow cytometry using FITC channel. As a negative control, P-gp inhibitor Verapamil (Ozon, Tolyatti, Russia) was added to the samples to a final concentration 10 µM 45 min before the Rh123.

### 4.10. Statistical Analysis

Data are expressed as mean, ±SD. Statistical analysis was performed using Student’s *t*-test. Differences were considered significant at *p* values less than 0.05.

## 5. Conclusions

We have shown that overexpression of EGFR in MCF7 cells transforms the adhesive cell culture to a 3D culture. In contrast to MCF7^wt^ spheroids, these MCF7-EGFR spheroids were not enriched by CSC-like cells or P-gp active cells. Simultaneous partial loss of HER3 receptor in MCF7^wt^ spheroids and in MCF7-EGFR spheroids indicates a crucial role of spheroid-associated hypoxia in this process rather than EGFR overexpression.

## Figures and Tables

**Figure 1 ijms-22-12937-f001:**
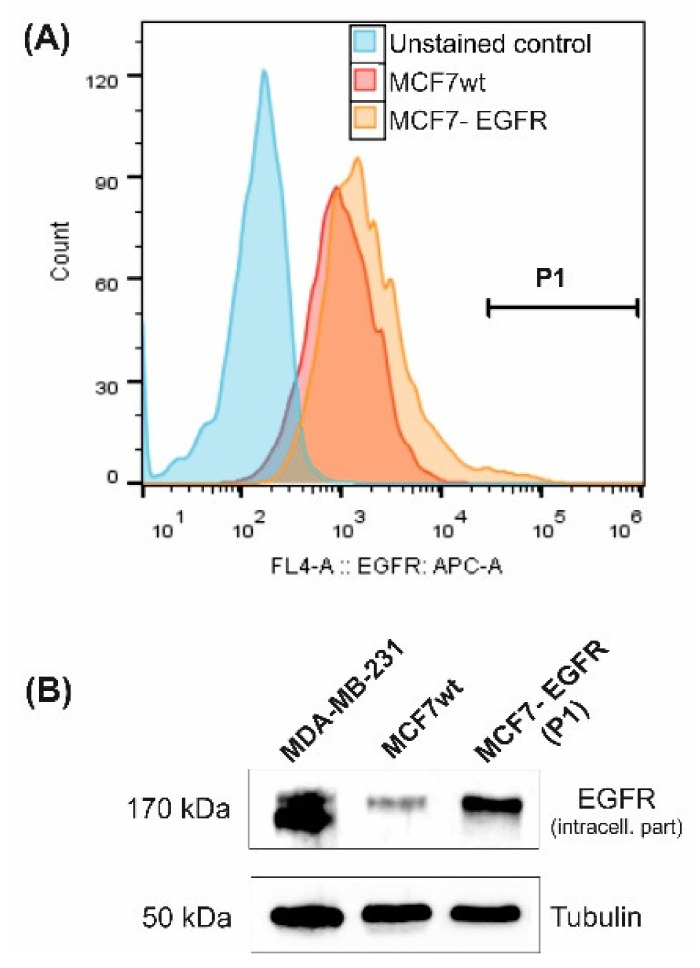
Characterization of MCF7-derived MCF7-EGFR cells. (**A**) Flow cytometry analysis of surface EGFR expression. Cells from gate P1 were sorted and cultivated as MCF7-EGFR cells. (**B**) Western blot of total cellular EGFR (typical image). Tubulin was used as a loading control.

**Figure 2 ijms-22-12937-f002:**
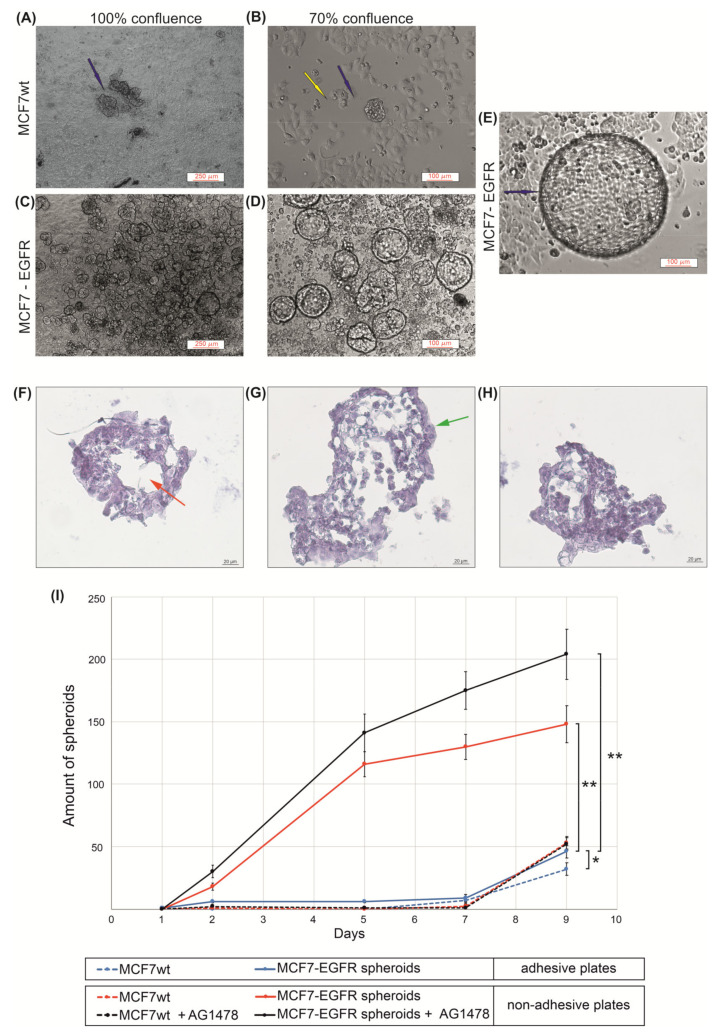
The measuring of sphere-forming ability of MCFwt and MCF7-EGFR cells. (**A**–**E**) Typical images of spheroids (phase contrast). Images of MCF7-EGFR cells were made on days 11 and 14 after the FACS sorting; (**F**–**H**) Typical images of microstructure of spheroids. Hematoxylin and eosin staining of histological sections. The red asterisk indicates the necrotic core of the spheroid; the green asterisk indicates the proliferative zone of the spheroid; (**I**) Growth kinetics of spheroids in adhesive and non-adhesive plates. Cells were counted in the individual wells of 96-well plates, and the amount of spheroid per well is indicated. The final concentration of AG1478 in wells was 10 µM. Data presented as mean value ± SD of three independent experiments; * *p* < 0.05 and ** *p* < 0.01.

**Figure 3 ijms-22-12937-f003:**
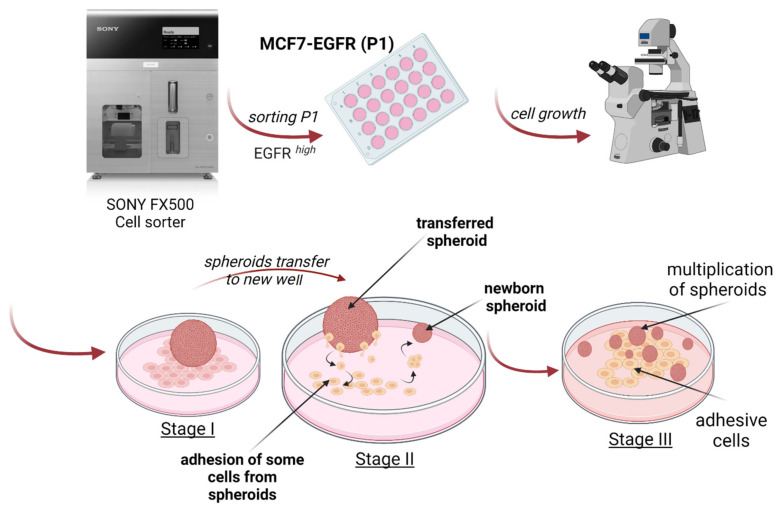
Schematic representation of MCF7-EGFR spheroids generation and stages of cultivation.

**Figure 4 ijms-22-12937-f004:**
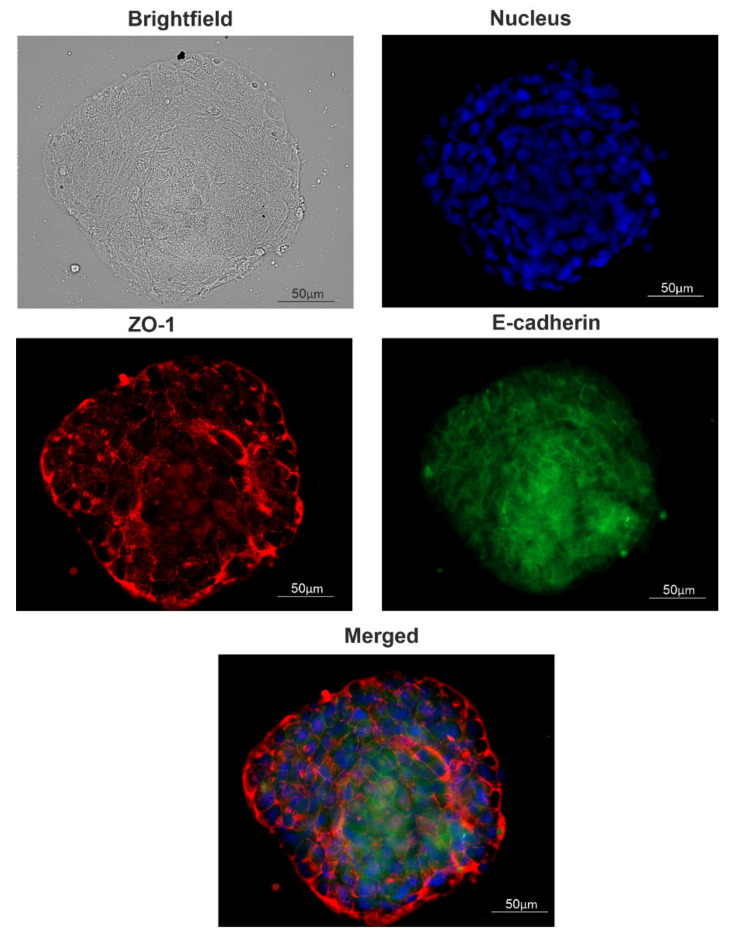
Tight junction proteins distribution in spheroids. A typical image of a spheroid. The presence of junction markers zona occludens-1 (ZO-1) and E-cadherin were confirmed via immunofluorescent staining: red, ZO-1; green, E-cadherin; blue, nuclei (DAPI).

**Figure 5 ijms-22-12937-f005:**
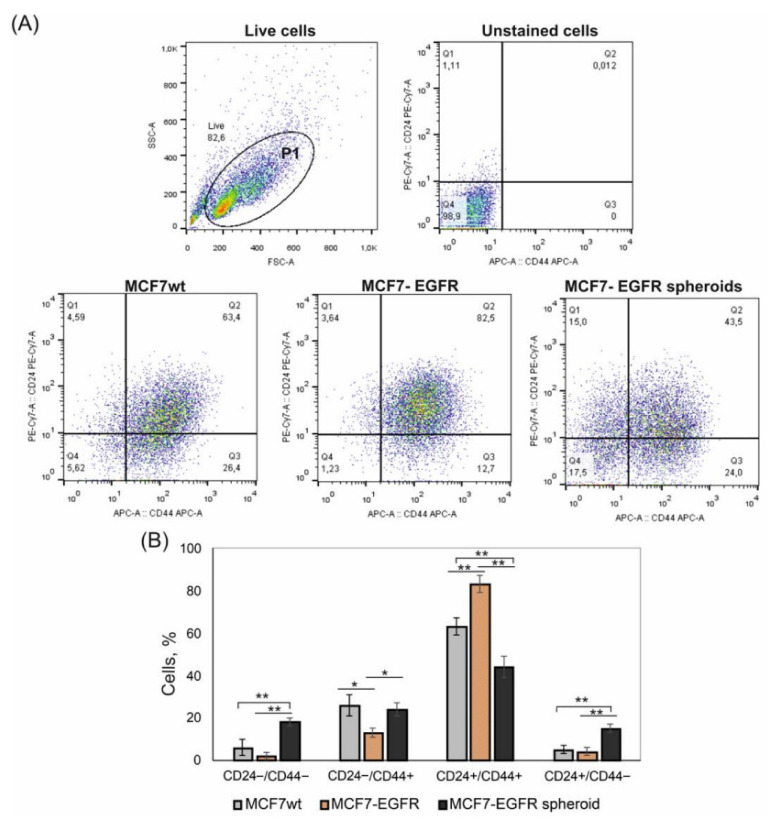
Flow cytometry analysis of CD44 and CD24 markers in MCF7 and MCF7-derived cells. The initial gating (P1) was made to exclude cell debris. Analysis was made on day 4 of cultivation. (**A**) Typical images of analysis. (**B**) Relative contribution of the CD44^+^/CD24^−^ and CD44^+^/CD24^+^ subpopulations in cell cultures. Bar graph showing the percentage of cells with indicated phenotype detected by flow cytometry. Statistical analysis included the results of three independent experiments (mean ± SD). The difference between the two groups was statistically significant at *p* < 0.05 (*) and at *p* < 0.01 (**).

**Figure 6 ijms-22-12937-f006:**
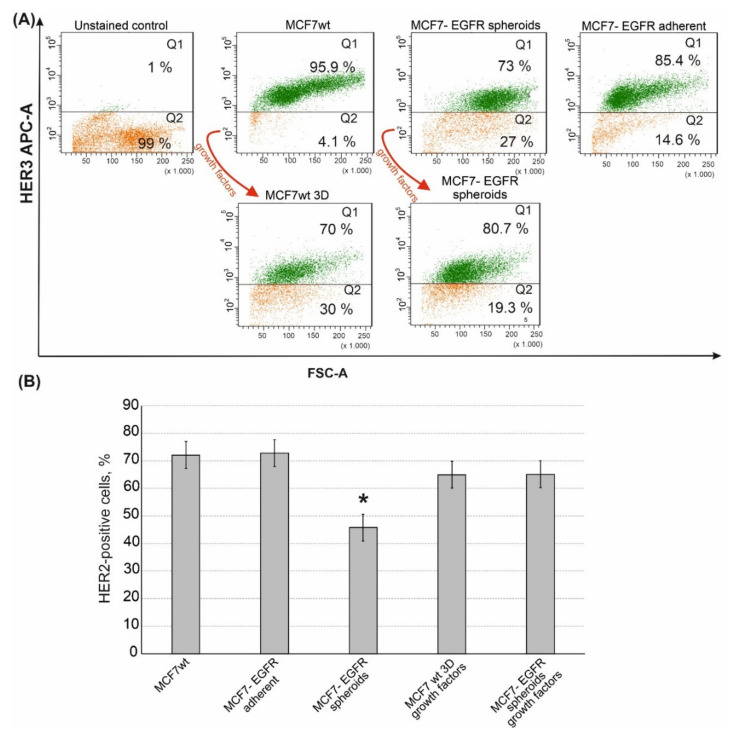
Flow cytometry analysis of HER3 and HER2 receptors in MCF7 and MCF7-derived cells. The growth factors cocktail is described in the Materials and Methods Section. (**A**) Typical images of analysis of HER3 receptor. (**B**) Quantification of the HER2-positive populations that were identified by flow cytometry with anti-HER2 monoclonal antibodies. Data presented as mean % of HER2-positive cells ± SD of three independent experiments. The differences with MCF7^wt^ were significant with * *p* < 0.05 (ANOVA test).

**Figure 7 ijms-22-12937-f007:**
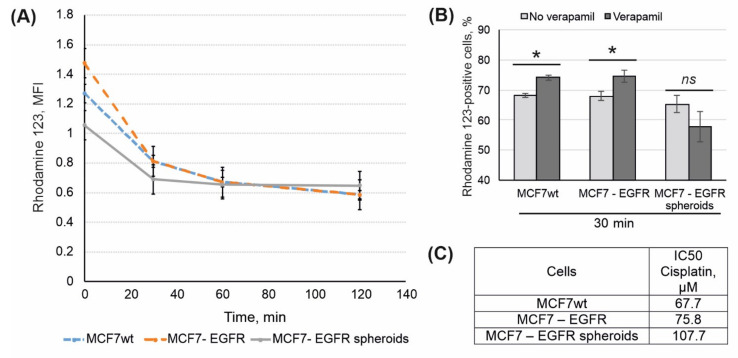
Analysis of drug resistance in MCF7^wt^ and MCF7-derived cells. (**A**) Kinetic of rhodamine 123 efflux in MCF7^wt^ and MCF7-derived cells. (**B**) Data presented as percent of Rh123-positive cells, that based on mean MFI of Rh123 signal. SD was calculated in four independent experiments. Verapamil (10 µM) was added to the samples for the efflux specificity control. (**C**) IC50 values for cisplatin were calculated using MTT data (48 h after the treatment). The differences between the two groups were significant with * *p* < 0.05 or ns, non-significant (Student’s *t*-test).

## Data Availability

Data are available in this manuscript or from authors upon reasonable request.

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
