# Peer review of "EGFR Transgene Stimulates Spontaneous Formation of MCF7 Breast Cancer Cells Spheroids with Partly Loss of HER3 Receptor"

_ijms, 2021, doi:10.3390/ijms222312937_

Round 1

Reviewer 1 Report

The manuscript by Troitskaya et al., investigates the effect of EGFR overexpression on spheroid forming in MCF7 cells. The findings propose overexpression of EGFR as a strategy to obtain a more suitable 3d model for studying breast cancer. The following aspects need to be addressed/improved:

  1. The authors need to provide a better characterization of the spheroids (e.g. expression of apical and basolateral markers, expression of tight junction proteins by fluorescence microscopy).
  2. A control analyzing spheroid formation in MCF7EGFR cells in the presence of an EGFR inhibitor should be performed.
  3. Figure 2. Please specify the culture time in the figure legend.
  4. The title of figures S1 and S2 is missing.
  5. Lines 123-125. The paragraph is not clear and does not fully reproduce the aim of the experiments. Please reformulate.
  6. Figure 5. The experiment of Rhodamine 123 efflux must be repeated in the presence of a P-gp inhibitor (e.g. PSC833) in order to confirm the specificity of the efflux.
  7. Section 4.2. Please specify how the spheroids were counted. Was any image analysis software used? Which was the threshold to define a positive count as spheroid?
  8. Section 4.6, line 313. Please, provide references for the standard protocols used for the quantification of rhodamine123 efflux.
  9. Lines 326 and 327. A comparison between P-gp efflux in MCF7wt spheroids and MCF7EGFR spheroids is made. However, no data on the efflux of P-gp efflux in MCF7wt spheroids is presented in the manuscript (Fig 5 presents only MCF7wt, MCF7EGFR and MCF7EGFR spheroids). Please, check.
  10. Please, discuss the mechanisms which could lead to the formation of 3d structures even in the absence of an extracellular matrix.
  11. The manuscript needs to be thoroughly checked for grammar and syntax mistakes.

Author Response

Reviewer 1

The manuscript by Troitskaya et al., investigates the effect of EGFR overexpression on spheroid forming in MCF7 cells. The findings propose overexpression of EGFR as a strategy to obtain a more suitable 3d model for studying breast cancer. The following aspects need to be addressed/improved.

Answer

We thank the reviewer for the positive feedback on our manuscript.

Comment 1. The authors need to provide a better characterization of the spheroids (e.g. expression of apical and basolateral markers, expression of tight junction proteins by fluorescence microscopy).

Answer

We did our best to address the critics of the reviewer. We added experimental data on morphological analysis of hematoxylin and eosin stained sections of spheroids (See Fig. 2F-H). These experiments required our significant efforts, and the optimization of the conditions for samples preparation to be quite a difficult task.

Unfortunately, due to the size of spheroids it is not possible to obtain a sufficient number of high-quality preparations to test the conditions of staining with antibodies in the near future, but we continue to work with this task.

Comment  2. A control analyzing spheroid formation in MCF7EGFR cells in the presence of an EGFR inhibitor should be performed.

Answer

In the current version of the manuscript we added experimental data of spheroid formation in MCF7EGFR cells with AG1478, an EGFR inhibitor (See Lines 135-142).

Comment 3. Figure 2. Please specify the culture time in the figure legend.

Answer

In the current version of the manuscript we indicated culture time in figure legend for Fig. 2. (See Lines 110-112) Moreover, since such a question arose, we have added a scheme of the experiment explaining the procedure of spheroids passing and cultivation (see new Fig. 3).

Comment 4. The title of figures S1 and S2 is missing.

Answer

We apologize that the tittles of figure S1 and S2 were missing during submission. Tittles were included in the new version of the manuscript.

Comment 5. Lines 123-125. The paragraph is not clear and does not fully reproduce the aim of the experiments. Please reformulate.

Answer

In the current version of the manuscript we reformulated the text (See Lines  145-147).

Comment 6. Figure 5. The experiment of Rhodamine 123 efflux must be repeated in the presence of a P-gp inhibitor (e.g. PSC833) in order to confirm the specificity of the efflux.

Answer

We thank the reviewer for these comments. In the current version of the manuscript we added experimental data on Rhodamine 123 efflux with P-gp inhibitor Verapamil (See Lines 211-213; 395-397; Fig. 6B).

Comment 7. Section 4.2. Please specify how the spheroids were counted. Was any image analysis software used? Which was the threshold to define a positive count as spheroid?

Answer

In the current version of the manuscript we added information on how the spheroids were counted (See Lines 332-334).

Comment 8. Section 4.6, line 313. Please, provide references for the standard protocols used for the quantification of rhodamine123 efflux.

Answer

We provided reference for the standard protocols used for the quantification of rhodamine123 efflux (See Ref. 41).

Comment 9. Lines 326 and 327. A comparison between P-gp efflux in MCF7wt spheroids and MCF7EGFR spheroids is made. However, no data on the efflux of P-gp efflux in MCF7wt spheroids is presented in the manuscript (Fig 5 presents only MCF7wt, MCF7EGFR and MCF7EGFR spheroids). Please, check.

Answer

We thank the reviewer for this remark, but would like to justify our vision. The aim of this experiment was to evaluate 1) the effect of EGFR expression on P-gp activity when comparing MCF7wt and MCF7EGFR, and 2) to evaluate P-gp activity between MCF7EGFR adherent and MCF7EGFR spheroids. We believe that the data presented allow us to make such a comparison.

Comment 10. Please, discuss the mechanisms which could lead to the formation of 3d structures even in the absence of an extracellular matrix.

Answer

In the current version of the manuscript we added speculations concerning to why these cells can exist as 3D without a matrix (See Lines 280-285).

Comment 11. The manuscript needs to be thoroughly checked for grammar and syntax mistakes.

Answer

Current version of the manuscript was proofed by native English speaker.

Reviewer 2 Report

Troitskaya et al reports data over the characteristics of 3D growth of breast cancer cells MCF7. They have engineered the MCF7 cells with overexpression of EGFR and studied the formation of spheroids. The find that increased EGFR expression correlates with increased 3D spheroid formation, but with lower expression of other ErbB receptors family members HER3 and HER2. Authors reports an increase of CD24-/CD44- cells, which might explain the higher number of formed spheroids. Although the data presented are correctly interpreted, overall the study missed major scientific rationale and justified models.

Specific comments:

“MCF7-EGFR cells have assembled in spheroids very fast and grew predominantly as 3D suspension culture with no special coating plastic, growth supplements or additional matrixes.” Have the authors compared also MCF7 cells infected with a control vector? Viral transductions sometimes can alter targeted cells, thus a control is needed.

“To ensure that high EGFR expression sufficient to transform the adhesive cell culture to spheroids we constructed the similar EGFR high cells of human embryonic kidney cells HEK293.“ Here the author cannot conclude that only EGFR is responsible for increased spheroid formation, as HEK293 are not cancer cells, well different than MCF7. One should better compare with ‘normal’ mammary cells as well other breast cancer cell lines or simply reduce the EGFR expression in the engineered MCF-7-EGFR cells. Immortalized breast cells MCF10A can form 3D structures, it would be interesting to know whether the removal of the ErbB receptors affects this process.

“We compared the representation of CD44 and CD24 markers on MCF7wt cells, MCF7- EGFR adherent cells, and MCF7-EGFR spheroids. EGFR transgene didn’t stimulate the increase of CD24-/low/CD44+ steam-like cells population. The content of these CD24-/low/CD44+ cells was approximately equal in the parental MCF7wt cells and spheroids. The main difference concerned the CD24-/low/CD44- population, which appeared in spheroids (Figure 3, quadrant Q1). A number of studies describe CD24-/low/CD44- cells as a population with stem plasticity which can realizes transition into CD24-/low/CD44+ cells.” In Figure 3 there is no quantification of the three independent experiments for the CD24/CD44 stainings. The differences between CD24-/CD44+ and double positive cells are not mentioned. In the next figure (only second panel) quantification is presented, but data are plotted with MFI and not over the number of events, is there any reason for it?

 “For understanding if the transformation to spheroids leads to the loss of HER3 we added sphere-promoting growth factors cocktail to the MCF7wt cells” This has no explanation.

Author Response

Reviewer 2

Troitskaya et al reports data over the characteristics of 3D growth of breast cancer cells MCF7. They have engineered the MCF7 cells with overexpression of EGFR and studied the formation of spheroids. The find that increased EGFR expression correlates with increased 3D spheroid formation, but with lower expression of other ErbB receptors family members HER3 and HER2. Authors reports an increase of CD24-/CD44- cells, which might explain the higher number of formed spheroids. Although the data presented are correctly interpreted, overall the study missed major scientific rationale and justified models.

Answer

We thank the reviewer for the high appreciation of our work.

Comment 1

“MCF7-EGFR cells have assembled in spheroids very fast and grew predominantly as 3D suspension culture with no special coating plastic, growth supplements or additional matrixes.” Have the authors compared also MCF7 cells infected with a control vector? Viral transductions sometimes can alter targeted cells, thus a control is needed.

Answer

We thank the reviewer for this perfect comment concerning control vector. Indeed, at the initial stages of the study we observed the growth of the culture infected with the control vector, and found no change in the frequency of spheroid formation compared with MCF7wt cells without transfection. We added this information to the current version of the manuscript (See Lines 106-107, 312, 315).

Comment 2 “To ensure that high EGFR expression sufficient to transform the adhesive cell culture to spheroids we constructed the similar EGFR high cells of human embryonic kidney cells HEK293“ Here the author cannot conclude that only EGFR is responsible for increased spheroid formation, as HEK293 are not cancer cells, well different than MCF7. One should better compare with ‘normal’ mammary cells as well other breast cancer cell lines or simply reduce the EGFR expression in the engineered MCF-7-EGFR cells. Immortalized breast cells MCF10A can form 3D structures, it would be interesting to know whether the removal of the ErbB receptors affects this process.

Answer

We agree with the reviewer that this aspect is more complex than it seems at first glance. However, would like to emphasize the logic of our experiments below:

2) HEK293 is true EGFRlow cell culture. There are few such cultures available in cell collections.

3) Since we did not see sphere formation in HEKEGFR culture, it is most reasonable to draw the conclusion we presented in the paper: " The findings indicate that the sphere-forming ability of MCF7-EGFR originated from the typical features of parental MCF7 cell." (See lines 149-151)

Nevertheless, we agree that it is more appropriate to continue working toward knockdown EGFR expression in MCF10A. We will plan such experiments in our next work, as we cannot promptly obtain MCF10A culture for use because of the Covid19 pandemic various restrictions.

Comment 3

 “We compared the representation of CD44 and CD24 markers on MCF7wt cells, MCF7- EGFR adherent cells, and MCF7-EGFR spheroids. EGFR transgene didn’t stimulate the increase of CD24-/low/CD44+ steam-like cells population. The content of these CD24-/low/CD44+ cells was approximately equal in the parental MCF7wt cells and spheroids. The main difference concerned the CD24-/low/CD44- population, which appeared in spheroids (Figure 3, quadrant Q1). A number of studies describe CD24-/low/CD44- cells as a population with stem plasticity which can realizes transition into CD24-/low/CD44+ cells.” In Figure 3 there is no quantification of the three independent experiments for the CD24/CD44 stainings. The differences between CD24-/CD44+ and double positive cells are not mentioned.

Answer

We thank the reviewer for these comments. In the current version of the manuscript, we modified Figure 3 according to the reviewer's comment (See quantification on Fig.3B).  The differences between CD24-/CD44+ and double positive cells were also mentioned (See Lines 168-170).

Comment 3

In the next figure (only second panel) quantification is presented, but data are plotted with MFI and not over the number of events, is there any reason for it?

Answer

As requested by the reviewer, in the new version the histograms are plotted based on (%) of HER2+ cells (See Fig. 5B).

Comment 4

 “For understanding if the transformation to spheroids leads to the loss of HER3 we added sphere-promoting growth factors cocktail to the MCF7wt cells” This has no explanation.

Answer

We thank the reviewer for these comments. Please, see new explanation, Lines 192-196

Round 2

Reviewer 1 Report

Although some of my concerns were addressed in this revised version, the manuscript still fails to answer to critical issues:

1- Characterization of MCF-7 spheroids. The authors provided hematoxylin-eosin stainings of spheroids. Characterization by immunocytochemistry is absolutely necessary for these type of 3d structures. Noteworthy, this type of technique has been already published in the literature for similar types of 3d structures and, therefore, should be accesible for the authors (e.g. Mokhtari et al., Cancers (Basel) 2021; Boyer et al., Breast Cancer Res Treat, 2021, etc. )

2- The authors added information of the drug resistance exhibited by the spheroids. Nevertheless, the data and its interpretation are significantly flawed. First of all, no statistical analysis was performed for any of the experimental conditions (Fig 6 A, B, C). Furthermore, lines 221-222 claim a higher rate of efflux for Rh123 in MCF7-EGFR cells. However, this is not visible evidenced by the data. In addition, no clear effect of verapamil is observed. Finally, the authors used cisplatin as example of cytostatic drug and claim an increased resistance in MCF7-EGFR cells and spheroids. The selection of this drug is not fully clear, specially considering that cisplatin is not a substrate of P-gp. 

Author Response

Although some of my concerns were addressed in this revised version, the manuscript still fails to answer to critical issues:

Comment 1.  Characterization of MCF-7 spheroids. The authors provided hematoxylin-eosin stainings of spheroids. Characterization by immunocytochemistry is absolutely necessary for these type of 3d structures. Noteworthy, this type of technique has been already published in the literature for similar types of 3d structures and, therefore, should be accesible for the authors (e.g. Mokhtari et al., Cancers (Basel) 2021; Boyer et al., Breast Cancer Res Treat, 2021, etc. )

Answer

We are infinitely grateful to the reviewer for recommending the methodology of immunostaining to characterize spheroids. As recommended by the reviewer, immunostaining was performed and Figure 4 added to the manuscript (See Fig. 4, Lines 126-128, Lines 258-261 and Lines 358-374 in Methods).

Comment 2.  The authors added information of the drug resistance exhibited by the spheroids. Nevertheless, the data and its interpretation are significantly flawed. First of all, no statistical analysis was performed for any of the experimental conditions (Fig 6 A, B, C). Furthermore, lines 221-222 claim a higher rate of efflux for Rh123 in MCF7-EGFR cells. However, this is not visible evidenced by the data. In addition, no clear effect of verapamil is observed. Finally, the authors used cisplatin as example of cytostatic drug and claim an increased resistance in MCF7-EGFR cells and spheroids. The selection of this drug is not fully clear, specially considering that cisplatin is not a substrate of P-gp. 

Answer

As recommended by the reviewer, one more repeat of RH123 efflux experiment was performed, and the values of differences between the groups of Veropamil-/verapamil+ were recalculated (please, see Fig. 7B). The conclusion regarding drug resistance of cell lines has been restated (See Lines 230-231). Moreover, in accordance with the reviewer's comment, we excluded the conclusion "Functional assay of MDR activity with Rhodamine 123 revealed that P- glycoprotein transporter was more efficient in MCF7-EGFR adherent cells than in MCF7wt cells or in MCF7-EGFR spheroids" from the Abstract.

We agree with the reviewer's observation that cisplatin is not a characteristic substrate of P-gp.

Reviewer 2 Report

Authors have replied and adapted the manuscript, which is now suitable for publication.

Author Response

We thank the reviewer for the positive feedback on our manuscript.

Round 3

Reviewer 1 Report

The authors provided a significantly improved version.

Before publication the authors must revise the statistical test used in figure 7. 

The authors state: "The differences between two groups were significant with *p<0.05 or ns-non significant (ANOVA test)". It has to be noted that the ANOVA-test is not suitable to assess for differences between two experimental groups. Data must be analized using a proper statistical test for two experimental groups. 

Author Response

We thank the reviewer for the remark. A correction has been made.  Comparison between the two groups was performed using Student's t-test (See Line 228).

Reviewer 2 Report

Manuscript is sufficiently revised.

Author Response

--